# Epidemiology of thalassemia among the hill tribe population in Thailand

**Tawatchai Apidechkul** [1,2]*, **Fartima Yeemard**[1], **Chalitar Chomchoei**[3], **Panupong Upala**[1], **Ratipark Tamornpark**[1]

**1** Center of Excellence for The Hill tribe Health Research, Mae Fah Luang University, Chiang Rai, Thailand, **2** School of Health Science, Mae Fah Luang University, Chiang Rai, Thailand, **3** Chulabhorn Royal Academy, Bangkok, Thailand

* Tawatchai.api@mfu.ac.th

## Abstract

### Background

Thalassemia is a severe disease that occurs due to abnormalities in hemoglobin genes. Various genetic factors in different populations lead to different clinical manifestations of thalassemia disease, particularly among people who have a long history of migration and who have married among tribes, such as the hill tribe people in Thailand. This genetic epidemiological study aimed to estimate the prevalence of various forms of thalassemia among the six main hill tribe populations in Thailand.

### Methods

A cross-sectional study was conducted to obtain information and blood specimens from school children belonging to one of the six main hill tribes in Thailand: Akha, Lau, Hmong, Yao, Karen, and Lisu. Hill tribe children who were attending grades 4–6 in 13 selected schools in Chiang Rai Province, Thailand, were invited to participate in the study. A validated questionnaire and 3 mL blood specimens were collected after obtaining information consent forms from both the children and their parents on a voluntary basis. A complete blood count (CBC) was performed, followed by osmotic fragility (OF) and dichlorophenol indophenol precipitation (DCIP) tests to screen for thalassemia. High-performance liquid chromatography (HPLC) and real-time quantitative polymerase chain reaction (qPCR) were used to identify hemoglobin type and α-thalassemia, respectively. A t-test, chi-square and logistic regression were used to detect the associations between variables at the significance level of α = 0.05.

### Results

A total of 1,200 participants from 6 different tribes were recruited for the study; 50.0% were males, and 67.3% were aged 11–12 years. The overall prevalence of thalassemia carriers according to the screening tests was 9.8% (117 of 1,200). Among the cases, 83 were A2A (59 cases were α-thalassemia 1 carrier or α-thalassemia 2 carrier or homozygous α-thalassemia 2, and 24 cases were β-thalassemia trait with or without α-thalassemia); 1 case was

**Funding:** TA was granted from The National Research Council of Thailand (NRCT) and Mae Fah

Luang University to support grant(Grant No.23/2561), but they did not involve to the project.

**Competing interests:** The authors have declared that no competing interests exist.

EE (homozygous Hb E with or without α-thalassemia); 31 cases were EA (30 cases were the Hb E trait, and 1 case was Hb E trait with or without α-thalassemia); 1 case was A2A Bart's H (Hb H disease α-thalassemia 1/α-thalassemia 2); and 1 case was A2A with abnormal Hb. The prevalence of the α-thalassemia 1 trait among the hill tribe population was 2.5%. The greatest prevalence of the α-thalassemia 1 trait was found in the Karen (3.0%) and Hmong (3.0%) tribes.

## Conclusions

The prevalence of some forms of thalassemia in the hill tribe population is higher than that in the Thai and other populations. Effective and available thalassemia screening tests, including essential information to protect the next generation through the specific counseling clinic, are crucial, particularly due to increasing marriages within these populations.

## Introduction

Thalassemia is a major inherited blood disorder caused by insufficient or nonfunctional hemoglobin [1]. Hemoglobin (Hb) is an important protein in red blood cells (RBCs) [2,3]. There are different forms of abnormal human red blood cells, which are indicated by the various levels of hemoglobin severity [3,4]. The Hb protein structure consists of 4 subunits, with each unit comprising heme and globin chains. There are two groups of globin chain: an α-like subunit, which consists of 141 amino acids (α-globin chain and ζ-chain), and a β-like subunit, which consists of 146 amino acids (β-globin chain, γ-globin chain, and δ-globin chain). Thalassemia is an abnormality in Hb in either quantitative or/qualitative aspects. Quantitative aspects result from aberrant expression of one of the globin chains: if expression of α-globin is reduced or absent, the disease is called "α-thalassemia"; if β-globin expression is reduced or absent, it is called "β-thalassemia". Qualitative aspects lead to abnormal Hb, which mainly occurs from point mutations. There are two forms that are often found in the Thai population: Hb E, which is caused by missense mutation of the β-globin gene at codon 26 from GAG to AAG; and Hb Constant Spring (Hb CS), which is caused by mutation of the α-global gene ($\alpha^{cs}$) at codon 142 from TAA (stop codon) to CAA [4].

Today, thalassemia is widely found with different variants in different populations [2]. In 2018, the World Health Organization (WHO) reported that at least 5.2% of individuals worldwide were thalassemia carriers, that approximately 1.1% of couples worldwide were at risk of having children with a hemoglobin disorder, and that 2.7/1,000 conceptions were affected [5]. Most children born with thalassemia in high-income countries survive with a chronic disorder, whereas children in developing countries die before 5 years of age [5]. In 2018, the Ministry of Public Health in Thailand reported that approximately 30–40% of Thai individuals (18–24 million people) were carriers of thalassemia or hemoglobin disorders, with more than 12,000 infants born with thalassemia every year [6]. Moreover, more than 600,000 thalassemia patients in Thailand are covered by the health system and require regular treatment, including blood transfusion [7]. These patients consume a large amount of medical resources for treatment and care. However, all medical expenses are covered by the government under the Thailand Universal Coverage Scheme (USC), which is required to cover approximately US $3,000 per year per person [8].

The United Nations reported that in early 2020, Thailand had a population of 69 million people [9]. In total, 4–5 million people were members of minority populations, such as hill

tribes and other stateless populations [10]. The hill tribe people comprise a group having migrated from the south of China to settle along the highland and border areas of Thailand-Myanmar-Republic of Laos [11]. There are six main groups: Akha, Lahu, Hmong, Yao, Karen, and Lisu [11]. The hill tribe people have their own languages, cultures, and lifestyle, which are different from those of native Thai people. With globalization and economic constraints, the hill tribe people have opened their villages to allow for work, education, and other businesses [12]. Many individuals in newer generations have started to marry individuals from other tribes, including native Thai. However, there is little available information regarding thalassemia epidemiology in hill tribe populations.

This study aimed to estimate the prevalence of thalassemia carriers and demonstrate the variations in hemoglobin types and magnitudes of differences in thalassemia genes among hill tribe populations in Thailand.

## Materials and methods

### Study design

A cross-sectional study was performed to collect data and blood specimens from hill tribe children who were attending grades 4–6.

### Study population

The target populations were hill tribe children who belonged to one of six main tribes: Akah, Lahu, Hmong, Yao, Karen, or Lisu.

### Study sample

The study participants were recruited from 13 of the 162 primary schools [13] located in the hill tribe villages in Chiang Rai Province, Thailand. All children who were attending grades 4 to 6 in selected schools and able to identify themselves as members of one of the six main hill tribes were invited to participate in the study.

### Sample size

The sample size was calculated to obtain the number required for estimating a prevalence with a specified level of confidence and precision. The outcome was the prevalence of various forms of thalassemia among the six main hill tribe populations in Thailand. The prevalence of thalassemia in a previous study was reported to be 13.0% [14]. The standard formula for a cross-sectional to estimate the sample size was applied, as follows [15]: n = [$Z^2_{\alpha/2}$ PQ]/$e^2$ ($Z_{\alpha/2}$ = 1.96, $P$ = 0.13 [15], $Q$ = 0.87, and e = 0.05), where $Z$ = value from the standard normal distribution corresponding to the desired confidence level (Z = 1.96 for 95% CI), $P$ is the expected true proportion, and $e$ is the desired precision; therefore, 173 participants from each tribe were needed. After adding an additional 10.0% for any error during the execution of the study, at least 192 participants from each tribe were required for analysis.

A validated questionnaire consisting of two parts was developed and used for data collection. In part one, 5 questions were used to collect personal information: age, sex, weight, height, and grade of school. In part two, 15 questions were used to ask questions regarding medical history and family history, such as the presence of medical conditions, history of blood transfusion, history of taking ferrous sulfate, and parents' tribe. A 3-mL blood specimen was collected from each participant.

## Data collection procedure

A total of 13 primary schools located in the hill tribe villages were randomly selected from 163 schools in Chiang Rai Province, Thailand [13]. Access to the schools for data collection was granted by the school directors. During a meeting with the school director, all essential information regarding the study was explained. Brief and essential information about the study was also provided to all children's parents five days before data and blood specimen collection. All children were attending grades 4–6, and the children of parents who voluntarily signed the informed consent form were eligible and invited to join the study. On the day of data collection, all children were provided information about the study again and asked regarding their willingness to provide informed consent. The questionnaire was completed by the students within 10 minutes. Afterwards, 3 mL of blood was drawn and collected in ethylenediaminetetraacetic acid (EDTA) tubes. The blood specimens were transferred to the Mae Fah Luang Medical Laboratory Center for analysis on the same day.

## Laboratory tests

A complete blood count (CBC) using the Sysmex/XN-550 was performed, followed by osmotic fragility (OF) [16] and dichlorophenol indophenol precipitation (DCIP) [17] screening tests. OF test was performed using KKU-OF reagent kit (Center for Research and Development of Medical Diagnosis Laboratory, Khon Kaen University, Khon Kaen Province, Thailand). A fresh EDTA-blood sample of 20 μL was used to add into the KKU-OF solution. Then, it was mixed by inventing, and placed the mixture at the room temperature for at least 15 minutes, before visual inspection. The interpretation was determined by observing the turbidity against the lines or letters; negative result was indicated by the clear solution, and positive result was indicated by turbidity. A positive quality control (mean corpuscular volume (MCV)>85fl, mean corpuscular hemoglobin (MCH) >28 pg, and Hb>13g/dL) and negative quality control (MCV<70 fl, MCH <25 pg, and Hb>10g/dL) were performed in parallel with unknown samples.

While, DCIP test was performed using KKU-DCIP-Clear reagent (Center for Research and Development of Medical Diagnosis Laboratory, Khon Kaen University, Khon Kaen Province, Thailand). The DCIP test required three reagents; the DCIP reagent (a 2 mL dark-blue solution form), the cleaning reagent (a dry powder form), and the diluent. The procedure started from a 20 μL EDTA blood sample and added 2 mL of DCIP reagent into the tube. Then, mixing was operated by a vortex mixer or inventing. Afterwards, a 15-minute incubation at the temperature room (37˚ C) was required before a 20-μL of cleaning solution was added and mixed. A tube was placed a few seconds or until the blue color disappeared at room temperature before determining the result. A clear color solution was negative, while turbidity was a positive result. The OF and DCIP tests were found a high sensitivity, specificity, and accuracy to detect α-thalassemia, β-thalassemia, and Hb-E while using in parallel test [18].

If either or both tests presented a positive result, a high-performance liquid chromatography (HPLC) was performed using VARIANT-II-hemoglobin testing system (BIO-RAD®, USA). Identifying β-thalassemia trait, we used the characteristics of Hb typing and also OF and DCIP tests according to the WHO guideline [19] and Thailand, Ministry of Public Health guideline [20]. The elution buffer 1 (Bio-Rad Laboratories, USA) was used to identify the hemoglobin type. Relative quantitative polymerase chain reaction (qPCR) was performed to identify α-thalassemia SEA and Thai deletion by using DMSc α-thal 1, which is produced by the Department of Medical Sciences, Ministry of Public Health, Thailand (Fig 1).

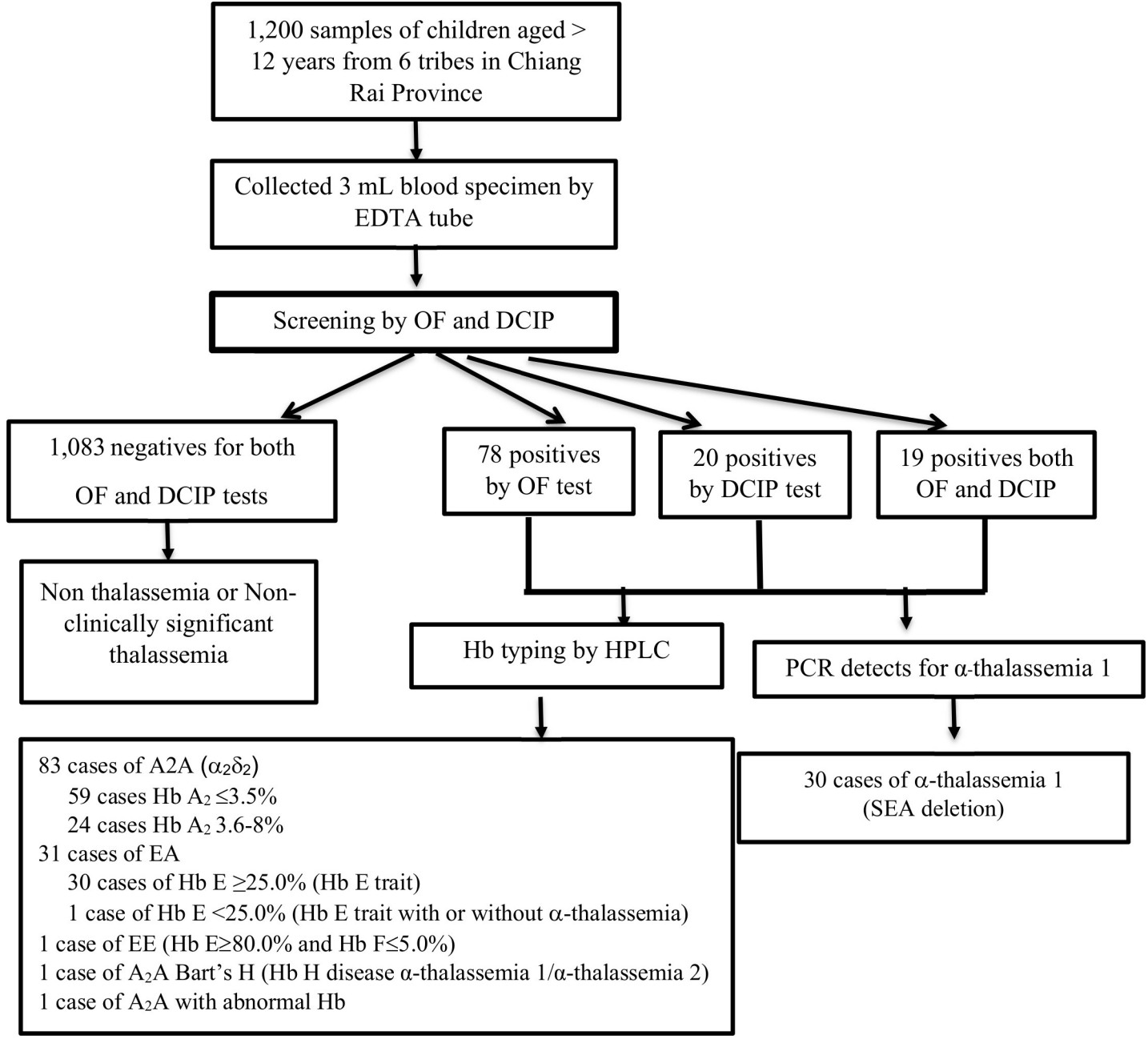

**Fig 1. Steps of data collection and laboratory analysis.**

## Statistical analysis

Data were doubled entered into an Excel sheet before being transferred into SPSS version 24 (SPSS, Chicago, IL) for analysis. The general characteristics of the participants are presented as the mean and standard deviation for continuous data; categorical data are described as percentages. A t-test and chi-square test were used for comparing means and proportions between groups, respectively. A logistic regression was used to detect the odds of having a positive for screening tests of thalassemia among the tribes.

## Ethical considerations

All study protocols were approved by the Research Ethics Committee of Chiang Rai Public Health Office (No. CRPPHO 14/2562). Before starting the interview, all participants were provided with all relevant and essential information. An informed consent form was obtained on a voluntary basis from both parents and children before starting the interviews. Moreover, all questionnaires were destroyed properly after being coded and entered into the Excel sheet, such that the data could not be traced back to any individual.

## Results

A total of 1,200 participants from 6 different tribes were recruited for the study; 50.0% were males, and 67.3% were aged 11–12 years. More than half of the participants were Christian (58.8%), all participants attended grades 4–6, and the participants were evenly distributed among the six tribes. The majority had 6 or fewer family members (Table 1).

**Table 1. General characteristics of the participants.**

| Characteristics | n | % |
|---|---:|---:|
| **Total** | **1,200** | **100.0** |
| **Sex** | | |
| Male | 600 | 50.0 |
| Female | 600 | 50.0 |
| **Age** (years) | | |
| 9–10 | 392 | 32.7 |
| 11–12 | 808 | 67.3 |
| *Mean = 11.03, Standard deviation (SD) = 0.84* | | |
| **Religion** | | |
| Buddhist | 495 | 41.2 |
| Christian | 705 | 58.8 |
| **Education** | | |
| Grade 4 | 391 | 32.6 |
| Grade 5 | 399 | 33.3 |
| Grade 6 | 410 | 34.2 |
| **Number of family members** | | |
| ≤ 6 | 975 | 81.2 |
| > 6 | 225 | 18.8 |
| **Number of siblings** | | |
| None | 184 | 15.3 |
| 1–2 | 667 | 55.6 |
| ≥ 3 | 349 | 29.1 |
| **Tribe** | | |
| Akha | 200 | 16.7 |
| Lahu | 200 | 16.7 |
| Hmong | 200 | 16.7 |
| Yao | 200 | 16.7 |
| Karen | 200 | 16.7 |
| Lisu | 200 | 16.7 |
| **Father's tribe** | | |
| Akha | 202 | 16.8 |

(*Continued*)

**Table 1.** (Continued)

| Characteristics | n | % |
|---|---|---|
| Lahu | 221 | 18.4 |
| Hmong | 194 | 16.2 |
| Yao | 194 | 16.2 |
| Karen | 195 | 16.3 |
| Lisu | 194 | 16.2 |
| **Mother's tribe** | | |
| Akha | 481 | 40.1 |
| Lahu | 239 | 19.9 |
| Hmong | 91 | 7.6 |
| Yao | 181 | 15.1 |
| Karen | 161 | 13.4 |
| Lisu | 47 | 3.9 |
| **Paternal grandfather's tribe** | | |
| Akha | 204 | 17.0 |
| Lahu | 188 | 15.7 |
| Hmong | 200 | 16.7 |
| Yao | 201 | 16.7 |
| Karen | 200 | 16.7 |
| Lisu | 207 | 17.3 |
| **Paternal grandmother's tribe** | | |
| Akha | 135 | 11.3 |
| Lahu | 191 | 15.9 |
| Hmong | 281 | 23.4 |
| Yao | 202 | 16.8 |
| Karen | 190 | 15.8 |
| Lisu | 201 | 16.8 |
| **Maternal grandfather's tribe** | | |
| Akha | 481 | 40.1 |
| Lahu | 239 | 19.9 |
| Hmong | 91 | 7.6 |
| Yao | 181 | 15.1 |
| Karen | 161 | 13.4 |
| Lisu | 47 | 3.9 |
| **Maternal grandmother's tribe** | | |
| Akha | 234 | 19.5 |
| Lahu | 136 | 11.3 |
| Hmong | 292 | 24.3 |
| Yao | 153 | 12.8 |
| Karen | 104 | 8.7 |
| Lisu | 281 | 23.4 |

According to medical history, only 1.8% reported having a medical condition, 1.1% had anemia, and a large proportion had taken ferrous sulfate (98.7%) (Table 2).

Comparisons of the mean corpuscular volume (MCV), mean corpuscular hemoglobin, hemoglobin (Hb), hematocrit (Hct), and red blood cells between those who were positive in either or both of the OF and DCIP tests and those who had negative results in both tests revealed significant differences (p-value = 0.010) (Table 3).

**Table 2. Medical history.**

| Medical history | n | % |
|---|---|---|
| **Medical condition** | | |
| No | 1,179 | 98.3 |
| Yes | 21 | 1.8 |
| *Allergy* | *14* | *1.2* |
| *Asthma* | *2* | *0.2* |
| *Peptic ulcer* | *5* | *0.4* |
| **Having anemia** | | |
| No | 1,187 | 98.9 |
| Yes | 13 | 1.1 |
| *Receiving blood transfusion* | *0* | *0.00* |
| **Having a family member receiving blood transfusion for anemia** | | |
| No | 1,200 | 100.0 |
| Yes | 0 | 0.0 |
| **Having relatives receiving blood transfusion for anemia** | | |
| No | 1,200 | 100.0 |
| Yes | 0 | 0.0 |
| **Having taken ferrous sulfate** | | |
| No | 15 | 1.3 |
| Yes | 1,185 | 98.7 |

The overall prevalence of thalassemia carriers according to the screening tests was 9.8%. The greatest prevalence of OF was found in Karen (13.0%), followed by Yao (9.0%) and Lahu (7.0%). Regarding the DCIP test, the highest prevalence was found in Hmong (6.0%), followed by Yao (4.0%). The proportions between positive and negative results of OF and DCIP tests were not statistically significant among the tribes. However, the differences in the proportions between sexes with regard to DCIP were significant (p-value = 0.002) (Table 4).

Regarding the proportions of OF- and DCIP-positive results in the one to two preceding generations, only maternal grandfathers and grandmothers presented significantly different results (p-value = 0.044) (Table 5).

Among either or both positives of OF and DCIP tests, 83 of 117 cases were A2A (32 cases had normal Hb typing, not rule out α-thalassemia, and 24 cases had the β-thalassemia trait with or without α-thalassemia). In addition, 1 case was EE (homozygous Hb E with or without α-thalassemia, 31 cases were EA (30 cases were Hb E trait, 1 case was Hb E trait with or without α-thalassemia), one case was A2A Bart's H (Hb H disease α-thalassemia 1/α-thalassemia 2), and one case was A2A with abnormal Hb (MCV = 73.4lf, positive OF, positive DCIP, 46.0%HbAo, 3.0%HbA2/E, 0.5%HbF, A2A abnormal Hb, and negative for alpha-thalassemia (SEA and Thai deletion).

**Table 3. Comparisons of mean MCV, MCH, Hb and Hct levels between thalassemia carriers and nonthalassemia carriers.**

| Biomarkers | Total (n = 1,200) | OF/DCIP positive (n = 117) | OF/DCIP negative (n = 1,083) | t-test | p-value |
|---|---|---|---|---|---|
| **MCV** | Mean = 80.86 SD = 6.89 | Mean = 67.79 SD = 8.33 | Mean = 82.27 SD = 4.97 | 18.10 | <0.001* |
| **MCH** | Mean = 26.47 SD = 2.40 | Mean = 21.48 SD = 2.90 | Mean = 27.01 SD = 1.59 | 19.75 | <0.001* |
| **Hb** | Mean = 13.02 SD = 0.96 | Mean = 11.77 SD = 1.01 | Mean = 13.16 SD = 0.85 | 14.25 | <0.001* |
| **Hct** | Mean = 39.81 SD = 2.92 | Mean = 37.17 SD = 3.02 | Mean = 40.09 SD = 2.76 | 10.39 | <0.001* |
| **RBC** | Mean = 4.94 SD = 0.43 | Mean = 5.53 SD = 0.58 | Mean = 4.88 SD = 0.36 | -11.79 | <0.001* |

* Significant level at $\alpha = 0.010$.

**Table 4. Comparisons of OF and DCIP tests by tribe and sex.**

| Test | Positive | | Negative | | $\chi^2$ | p-value |
|---|---|---|---|---|---|---|
| | n | % | n | % | | |
| **Total of OF and DCIP** | **117** | **9.8** | **1,083** | **90.2** | **N/A** | **N/A** |
| **OF test** | **96** | **8.0** | **1,104** | **92.0** | **N/A** | **N/A** |
| **Tribe** | | | | | 10.19 | 0.070 |
| *Akha* | *11* | *5.5* | *189* | *94.5* | | |
| *Lahu* | *14* | *7.0* | *186* | *93.0* | | |
| *Hmong* | *15* | *6.5* | *185* | *92.5* | | |
| *Yao* | *18* | *9.0* | *182* | *91.0* | | |
| *Karen* | *26* | *13.0* | *174* | *87.0* | | |
| *Lisu* | *12* | *6.0* | *188* | *94.0* | | |
| **Sex** | | | | | 0.55 | 0.458 |
| *Male* | *45* | *7.5* | *555* | *92.5* | | |
| *Female* | *52* | *8.7* | *548* | *91.3* | | |
| **DCIP test** | **39** | **3.3** | **1,161** | **96.7** | **N/A** | **N/A** |
| **Tribe** | | | | | 8.82 | 0.116 |
| *Akha* | *2* | *1.0* | *198* | *99.0* | | |
| *Lahu* | *6* | *3.0* | *194* | *97.0* | | |
| *Hmong* | *12* | *6.0* | *188* | *94.0* | | |
| *Yao* | *8* | *4.0* | *192* | *96.0* | | |
| *Karen* | *5* | *2.5* | *195* | *97.5* | | |
| *Lisu* | *6* | *3.0* | *194* | *97.0* | | |
| **Sex** | | | | | 9.56 | 0.002* |
| *Male* | *10* | *1.7* | *590* | *98.3* | | |
| *Female* | *29* | *4.8* | *571* | *95.2* | | |

* Significance level at $\alpha = 0.05$.

The prevalence of the α-thalassemia 1 trait (SEA deletion) among the hill tribes was 2.5% (2.3% in males and 2.7% in females) (Table 6).

According to the PCR method to identify α-thalassemia 1, the greatest proportion of patients with the α-thalassemia 1 trait (SEA deletion) was found in Karen (3.0%) and Hmong (3.0%) tribes. Regarding β-thalassemia detection, the greatest proportions were found in Yao (4.5%) and Karen (3.5%) tribes, whereas the greatest proportions of the Hb E trait were found in Hmong (4.5%) and Lisu (3.0%) tribes. One case of A2A Bart's H was found in a Lahu boy, one case of A2A with abnormal Hb was found in another Lahu boy, and one case of homozygous Hb E with or without α-thalassemia was found in a girl of Akha (Table 7).

In the logistic regression analysis to identify the associations between tribes and the OF and/or DCIP positives, it was found that Karen were more likely to have OF and/or DCIP positives than Akha (OR = 2.45, 95%CI = 1.20–4.98), in the overall model. While extracting into different sexes, Yao-males were more likely to have OF and/or DCIP positives than Akha-males (OR = 3.27, 95%CI = 1.02–10.53), and Karen-females were more likely to have OF and/or DCIP positives than Akha-males (OR = 5.25, 95%CI = 1.04–6.11). However, those who had same or different parents and grandparents were not found the association with OF and/or DCIP positives (Table 8).

**Table 5. Proportions of OF and DCIP results in preceding generations.**

| Pairs | Same-tribe | | Different-tribe | |
|---|---|---|---|---|
| | n | % | n | % |
| **Father's and mother's tribes** | | | | |
| *OF* | | | | |
| Positive | 23 | 6.3 | 73 | 8.8 |
| Negative | 343 | 93.7 | 761 | 91.2 |
| $\chi^2 = 2.10$, *p-value = 0.146* | | | | |
| *DCIP* | | | | |
| Positive | 9 | 2.5 | 30 | 3.6 |
| Negative | 357 | 97.3 | 804 | 96.4 |
| $\chi^2 = 1.04$, *p-value = 0.306* | | | | |
| **Both OF and DCIP positive (total = 19)** | 4 | N/A | 15 | N/A |
| **Paternal grandfather's and grandmother's tribes** | | | | |
| *OF* | | | | |
| Positive | 87 | 8.1 | 9 | 6.9 |
| Negative | 983 | 91.9 | 121 | 93.1 |
| $\chi^2 = 0.22$, *p-value = 0.631* | | | | |
| *DCIP* | | | | |
| Positive | 38 | 3.9 | 1 | 0.8 |
| Negative | 1,032 | 96.3 | 129 | 99.2 |
| $\chi^2 = 2.85$, *p-value = 0.091* | | | | |
| **Both OF and DCIP positive (total = 19)** | 18 | N/A | 1 | N/A |
| **Maternal grandfather's and grandmother's tribes** | | | | |
| *OF* | | | | |
| Positive | 13 | 13.3 | 83 | 7.5 |
| Negative | 85 | 86.7 | 1,019 | 92.5 |
| $\chi^2 = 4.01$, *p-value = 0.044*\* | | | | |
| *DCIP* | | | | |
| Positive | 4 | 4.1 | 35 | 3.2 |
| Negative | 94 | 95.9 | 1,067 | 96.8 |
| $\chi^2 = 0.23$, *p-value = 0.628* | | | | |
| **Both OF and DCIP positive (total = 19)** | 2 | N/A | 17 | N/A |

\* Significance level at $\alpha = 0.05$.

## Discussion

The overall prevalence of thalassemia carriers among the hill tribes according to the screening tests (OF and DCIP tests) was 9.8%; the greatest proportion of OF-positive results was found in Yao (9.0%), and the greatest proportion of DCIP-positive results was found in Hmong (6.0%). Females (4.8%) were significantly more likely to have a positive DCIP test than were males (1.7%). OF test positivity among children whose maternal grandfather and grandmother were in the same tribe was higher than among children whose maternal grandfather and grandmother were from different tribes. The major forms of Hb disorder was A2A (6.9%) and EA (2.6%). The prevalence of the α-thalassemia 1 trait among the hill tribe population was 2.5%.

A large study on the prevalence of thalassemia among 1,796 Thai women in northeastern Thailand reported that the highest prevalence of thalassemia in the country was 30.2%, with the highest prevalence of homozygous Hb E being 5.4%, the highest prevalence of the β-

**Table 6. All laboratory results.**

| Type | n | Male n (%) | Female n (%) |
|---|---|---|---|
| **Total** | **1,200 (100.0)** | **600 (50.0)** | **600 (50.0)** |
| **Screening test** | | | |
| Negative | 1,083 (90.2) | 552 (92.0) | 531 (88.5) |
| Positive | 117 (9.8) | 48 (8.0) | 69 (11.5) |
| *OF positive* | *78 (66.7)* | *38 (79.1)* | *40 (57.9)* |
| *DCIP positive* | *20 (17.1)* | *3 (6.3)* | *17 (24.6)* |
| *Both OF and DCIP positive* | *19 (16.2)* | *7 (14.4)* | *12 (17.3)* |
| **HPLC** (Hb typing) | | | |
| A2A | 83 (70.9) | 39 (81.2) | 44 (63.8) |
| *Hb A2 ≤3.5% (Normal Hb typing, not rule out α-thalassemia)* | *59 (71.1)* | *25 (42.4)* | *34 (57.6)* |
| *Hb A2 3.6–8% (β-thalassemia trait with or without α-thalassemia)* | *24 (28.9)* | *14 (58.3)* | *10 (41.7)* |
| EE | 1 (0.9) | 0 (0.0) | 1 (1.4) |
| *Hb E≥80.0% and Hb F ≤ 5.0% (Homozygous Hb E with or without α-thalassemia)* | *1 (100.0)* | *0 (0.0)* | *1 (100.0)* |
| EA | 31(26.4) | 7(14.6) | 24 (34.8) |
| *Hb E >25.0% (Hb E trait)* | *30 (96.8)* | *7 (23.3)* | *23 (76.7)* |
| *Hb E< 25.0% (Hb E trait with or without α-thalassemia)* | *1 (3.2)* | *0 (0.0)* | *1 (100.0)* |
| A2A Bart's H | 1 (0.9) | 1(2.1) | 0(0.0) |
| *Hb H disease α-thalassemia 1/α-thalassemia 2* | *1(100.0)* | *1 (100.0)* | *0 (0.0)* |
| A2A with abnormal Hb | 1(0.9) | 1(2.1) | 0(0.0) |
| *Suspected abnormal Hb* | *1(100.0)* | *1 (100.0)* | *0 (0.0)* |
| **PCR detects for α-thalassemia 1** | | | |
| Positive for α-thalassemia (SEA deletion) | 30 (100.0) | 14 (46.7) | 16 (53.3) |
| *α -thalassemia 1 trait* | *30 (100)* | *14 (46.7)* | *16 (53.3)* |

thalassemia trait being 0.6%, and the highest prevalence of the α-thalassemia 1 trait being 3.0% [21]. These results reflect that the prevalence of some types of thalassemia among the hill tribes is higher than that among Thai people, even those living in the areas with the highest prevalence of thalassemia.

**Table 7. Proportions of Hb typing and thalassemia carriers by tribe.**

| Type | Total n | Total % | Akha n | Akha % | Lahu n | Lahu % | Hmong n | Hmong % | Yao n | Yao % | Karen n | Karen % | Lisu n | Lisu % |
|---|---|---|---|---|---|---|---|---|---|---|---|---|---|---|
| **Hb typing** | 117 | 100.0 | 12 | 10.3 | 18 | 15.4 | 23 | 19.7 | 21 | 17.9 | 27 | 23.1 | 16 | 13.7 |
| A2A *(Normal Hb typing, not rule out α -thalassemia)* | 59 | 50.4 | 9 | 4.5 | 10 | 5.0 | 10 | 5.0 | 7 | 3.5 | 15 | 7.5 | 8 | 4.0 |
| A2A *(β-thalassemia trait with or without α -thalassemia)* | 24 | 2.1 | 1 | 0.5 | 1 | 0.5 | 4 | 2.0 | 9 | 4.5 | 7 | 3.5 | 2 | 1.0 |
| EE | 1 | 0.9 | 1 (F) | 0.5 | 0 | 0.0 | 0 | 0.0 | 0 | 0.0 | 0 | 0.0 | 0 | 0.0 |
| EA | 31 | 26.5 | 1 | 0.5 | 5 | 2.5 | 9 | 4.5 | 5 | 2.5 | 5 | 2.5 | 6 | 3.0 |
| A2A Bart's H | 1 | 0.9 | 0 | 0.0 | 1 (M) | 0.5 | 0 | 0.0 | 0 | 0.0 | 0 | 0.0 | 0 | 0.0 |
| A2A with abnormal Hb | 1 | 0.9 | 0 | 0.0 | 1 (M) | 0.5 | 0 | 0.0 | 0 | 0.0 | 0 | 0.0 | 0 | 0.0 |
| **α-thalassemia 1** | | | | | | | | | | | | | | |
| α-thalassemia 1 trait (SEA deletion) | 30 | 25.6 | 4 | 2.0 | 5 | 2.5 | 6 | 3.0 | 5 | 2.5 | 6 | 3.0 | 4 | 2.0 |

* F: Female, M: Male.

**Table 8. Logistic regression analysis in identifying the associations between the difference of tribes and OF and/or DCIP positives.**

| Tribe | Total n (%) | OF/DCIP | | OR | 95%CI | p-value |
|---|---|---|---|---|---|---|
| | | Positive n (%) | Negative n (%) | | | |
| **Total** | **1,200(100.0)** | **117 (9.8)** | **1,083(90.3)** | | | |
| Akha | 200 (16.7) | 12 (6.0) | 188 (94.0) | 1.00 | | |
| Lahu | 200 (16.7) | 18 (9.0) | 182 (91.0) | 1.55 | 0.73–3.31 | 0.258 |
| Lisu | 200 (16.7) | 16 (8.0) | 184 (92.0) | 1.36 | 0.63–2.96 | 0.435 |
| Hmong | 200 (16.7) | 23 (11.5) | 177 (88.5) | 2.04 | 0.98–4.21 | 0.055 |
| Yao | 200 (16.7) | 21 (10.5) | 179 (89.5) | 1.84 | 0.88–3.85 | 0.106 |
| Karen | 200 (16.7) | 27 (13.5) | 173 (86.5) | 2.45 | 1.20–4.98 | 0.014* |
| **Male** | **600 (100.0)** | **48 (8.0)** | **552 (92.0)** | | | |
| Akha | 100 (16.7) | 4 (4.0) | 96 (96.0) | 1.00 | | |
| Lahu | 100 (16.7) | 9 (9.0) | 91 (91.0) | 2.37 | 0.71–7.98 | 0.162 |
| Lisu | 100 (16.7) | 6 (6.0) | 94 (94.0) | 1.53 | 0.42–5.60 | 0.519 |
| Hmong | 100 (16.7) | 8 (8.0) | 92 (92.0) | 2.09 | 0.61–7.17 | 0.243 |
| Yao | 100 (16.7) | 12 (12.0) | 88 (88.0) | 3.27 | 1.02–10.52 | 0.047* |
| Karen | 100 (16.7) | 9 (9.0) | 91 (91.0) | 2.37 | 0.71–7.98 | 0.162 |
| **Female** | **600 (100.0)** | **69 (11.5)** | **531 (88.5)** | | | |
| Akha | 100 (16.7) | 8 (8.0) | 92 (92.0) | 1.00 | | |
| Lahu | 100 (16.7) | 9 (9.0) | 91 (91.0) | 1.14 | 0.42–3.08 | 0.800 |
| Lisu | 100 (16.7) | 10 (10.0) | 90 (90.0) | 1.28 | 0.48–3.38 | 0.622 |
| Hmong | 100 (16.7) | 15 (15.0) | 85 (85.0) | 2.01 | 0.82–5.03 | 0.126 |
| Yao | 100 (16.7) | 9 (9.0) | 91 (91.0) | 1.14 | 0.42–3.08 | 0.800 |
| Karen | 100 (16.7) | 18 (18.0) | 82 (82.0) | 2.52 | 1.04–6.11 | 0.040* |
| **Father/ Mother's tribe** | | | | | | |
| Same | 366 (30.5) | 28 (7.7) | 338 (92.3) | 0.69 | 0.45–1.08 | 0.106 |
| Different | 834 (69.5) | 89 (10.7) | 745 (89.3) | 1.00 | | |
| **Paternal grandfather/grandmother' s tribe** | | s | | | | |
| Same | 1,070 (89.2) | 108(10.1) | 962 (89.9) | 1.51 | 0.75–3.06 | 0.253 |
| Different | 130 (10.8) | 9 (6.9) | 121 (93.1) | 1.00 | | |
| **Maternal grandfather/grandmother's tribe** | | | | | | |
| Same | 98 (8.2) | 15 (15.3) | 83 (84.7) | 1.77 | 0.99–3.19 | 0.056 |
| Different | 1102 (91.8) | 102 (9.3) | 1,000(90.7) | 1.00 | | |

\* Significant level at α = 0.05.

The prevalence of some forms of thalassemia among the hill tribe people was lower than that in the Thai population, though the prevalence of other forms was higher than that in the Thai population. For instance, the prevalence of the Hb E trait among the hill tribe (2.5%) was lower than that among the Thai population. According to a study in northeastern Thailand, the most prevalent type of thalassemia is the Hb E trait (39.1%) [22]. Additionally, a study among people who did not present anemia or microcytosis in central Thailand did not detect the α-thalassemia 1 trait, but there was a 15.8% prevalence of the Hb E trait and a 0.6% prevalence of the β-thalassemia trait [23]. Another study among Thai blood donors found that the thalassemia trait prevalence was 21.1% [24]. In a study estimating the burden of α-thalassemia in Thailand using a comprehensive prevalence database of Southeast Asia, it was estimated that 3,595 (95% CI = 1,717–6,199) newborns will be born with severe α-thalassemia in Thailand in 2020 [25].

Several studies have sought to identify the prevalence of thalassemia among different non-Thai populations and have reported different prevalence rates for different types of thalassemia. For instance, a study among migrant workers in Thailand found the prevalence rates of the α-thalassemia 1 trait (1.8%) and the β-thalassemia trait (3.9%) to be highest among workers from Myanmar, whereas the prevalence of the Hb E trait was highest among workers from Laos [26]. A population-based study among Tai and Mon-Khmer ethnic groups in northern Thailand reported that 23.4% had the α-thalassemia trait and that 96.9% were heterozygous. The study also reported different prevalence of α-thalassemia traits among the tribes [27]. Apidechkul reported that several forms of Hb disorder were found among the Lahu women in Chiang Rai Province, Thailand; 4.3% were Hb E trait, 0.8% were β-thalassemia trait, and 0.8% were Hb E homozygous [28]. A study among the school children in northern Thailand, it was presented that the prevalence of β-thalassemia trait was 7.9% [29], which was greater than our study (2.0%).

The reported prevalence of thalassemia in different countries varies. A study in Pakistan found that early screening and detection of β-thalassemia could significantly reduce severe thalassemia in later generations in a population [30], and a study in Morocco presented a prevalence of α-thalassemia of 0.9% [31]. In Iraq, it was reported that the prevalence of thalassemia increased from 33.5/100,000 in 2010 to 37.1/100,000 in 2015, with β-thalassemia accounting for 73.9% of all cases [32]. Moreover, in the United States, the prevalence of thalassemia due to the immigration of people from different regions of the world has reportedly increased by 7.5% in recent decades [33]; 4.6% of Australian people were found to be carriers of α-thalassemia and β-thalassemia gene variants [34].

In our study, females were more likely to have thalassemia than males, and those whose maternal grandfather and grandmother were of the same tribe were significantly more likely to have thalassemia than those whose grandparents were of different tribes. Our study also found the α-thalassemia 1 trait to be presented at a higher rate in females than in males, with the opposite for the β-thalassemia trait. However, there are a few papers presenting these data.

A systematic review clearly demonstrated that thalassemia is associated with a large health care service burden in different countries [35]. Some forms of thalassemia require regular life-long blood transfusions [36]. A study in northern Thailand showed that thalassemia was one of the significant risk factors for cardiac iron overload and cardiovascular complications [37]. Moreover, among nontransfusion-dependent thalassemia (NTDT) patients, major complications are cholelithiasis (35.0%), abnormal liver function (29.0%), and extramedullary hematopoiesis (EMH) (25.0%) [38], and a prospective study reported that nontransfusion-dependent Hb E/β-thalassemia and α-thalassemia (Hb H disease) result in the development of several severe diseases, particularly gallstones and serious infections [39].

Research projects related to genetic diseases among the hill tribe population in Thailand are very challenging due to several factors, including language and transportation barriers [40]. Therefore, a clear understanding of the research project among participants, particularly their parents, is required. Moreover, in disseminating the results of testing, all information should be provided carefully during the counseling process. One more point of the limitation of the study, it.

## Conclusion

The hill tribe people in Thailand have with a high prevalence of thalassemia, particularly the α-thalassemia trait, the β-thalassemia trait, and the Hb E trait. Some tribes have a higher prevalence than others. A higher proportion the α-thalassemia trait and Hb E trait was found for females, with males exhibiting a higher incidence of the β-thalassemia trait. Moreover, those

with grandfathers and grandmothers on either the father or mother's side who were from the same tribe were at a greater risk of thalassemia than those whose grandparents' were from different tribes; conversely, those whose parents were from different tribes were at a greater risk of thalassemia than those whose parents were from the same tribe. To reduce the prevalence of all types of thalassemia among the hill tribe people in Thailand, a screening program should be available at all health care centers located in hill tribe villages. Moreover, essential information on thalassemia, including prevention measures, should be provided to these populations in local hill tribe languages. A specific counseling clinic including prenatal diagnosis for hemoglobin disorders should be initiated in all levels of health institutes located in the hill tribe villages in Thailand.

## Supporting information

**S1 Appendix. Questionnaire used in the study (Thai).**
(PDF)

**S2 Appendix. Questionnaire used in the study (English).**
(PDF)

**S3 Appendix. Data file of the study.**
(SAV)

## Acknowledgments

The authors thank all school directors and participants for providing all the essential information.

## Author Contributions

**Conceptualization:** Tawatchai Apidechkul, Chalitar Chomchoei.

**Data curation:** Tawatchai Apidechkul, Fartima Yeemard.

**Formal analysis:** Tawatchai Apidechkul.

**Funding acquisition:** Tawatchai Apidechkul.

**Investigation:** Tawatchai Apidechkul, Fartima Yeemard, Chalitar Chomchoei, Panupong Upala, Ratipark Tamornpark.

**Methodology:** Tawatchai Apidechkul, Ratipark Tamornpark.

**Project administration:** Tawatchai Apidechkul.

**Writing – original draft:** Tawatchai Apidechkul, Fartima Yeemard, Chalitar Chomchoei, Panupong Upala, Ratipark Tamornpark.

**Writing – review & editing:** Tawatchai Apidechkul, Fartima Yeemard, Chalitar Chomchoei, Panupong Upala, Ratipark Tamornpark.

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
