## [Decision Letter · Decision Letter 0]

23 Jul 2020

PONE-D-20-14674

Epi-genetics of thalassemia among the hill tribe population in Thailand

PLOS ONE

Dear Dr. Apidechkul,

Thank you for submitting your manuscript to PLOS ONE. After careful consideration, we feel that it has merit but does not fully meet PLOS ONE’s publication criteria as it currently stands. Therefore, we invite you to submit a revised version of the manuscript that addresses all the the points raised during the review process, as reported below.

We look forward to receiving your revised manuscript.

Kind regards,

Michela Grosso, Ph.D.

Academic Editor

PLOS ONE

Journal Requirements:

2. Please include additional information regarding the questionnaire used in the study and ensure that you have provided sufficient details that others could replicate the analyses. For instance, if you developed a questionnaire as part of this study and it is not under a copyright more restrictive than CC-BY, please include a copy, in both the original language and English, as Supporting Information.

3. Please state in your methods section the participant recruitment date.

4. We note you have included a table to which you do not refer in the text of your manuscript. Please ensure that you refer to Table 7 in your text; if accepted, production will need this reference to link the reader to the Table.

Reviewers' comments:

Reviewer's Responses to Questions

**Comments to the Author**

1. Is the manuscript technically sound, and do the data support the conclusions?

Reviewer #1: Partly

2. Has the statistical analysis been performed appropriately and rigorously? 

Reviewer #1: Yes

3. Have the authors made all data underlying the findings in their manuscript fully available?

Reviewer #1: Yes

4. Is the manuscript presented in an intelligible fashion and written in standard English?

Reviewer #1: No

5. Review Comments to the Author

Reviewer #1: The manuscript by Tawatchai Apidechkul et al. “Epi-genetics of thalassemia among the hill tribe population in Thailand” presents results of a genetic screening, aiming to estimate the prevalence of various forms of thalassemia among the six main hill tribe (minority) populations in Thailand. The issue of management of hemoglobinopathies patients is still apparent despite attempts to be controlled with prenatal testing and it is even nowadays one of the most common genetic diseases. The prevalence of all forms of thalassemia is high in countries with diverse ethnicity, high immigrant rate and in minority societies where consanguineous marriages are widely in practice. The early onset of the disease (few months after birth) and the application of expensive therapeutic protocols, including blood transfusions and chelating agents, adds an unendurable financial burden to healthcare systems.

This study appears to have been carefully performed and conclusions are relevant to the main scope of the study, the mandatory establishment of genetic counseling and prenatal diagnosis for hemoglobin disorders, especially in minority societies such as the hill tribe population in Thailand.

However, there are a number of major issues in the manuscript that need to be addressed before resubmission of revised version:

1) The term “epigenetics” in the title is irrelevant to the subject of this study. Epigenetics refers to biochemical alterations of the histone proteins or genome (DNA) or non-coding RNAs that differentially regulate gene expression levels.

2) The manuscript needs an extensive grammatical and syntactical editing. In its present form does not meet the criteria for publishing.

3) The authors need to be as precise as possible about how they did the genetic analysis:

a)The equation in the paragraph “sample size”, although its reference is given, its terms are not (even briefly) described.

b) The whole section of “materials and methods” is insufficient. The lab tests must be described and results obtained by each method performed, must be stated. Differences detected between lab methods and presented in the discussion section, must be explained and substantiated.

4) A figure presenting α-type and β-type globin genes with specific mutations highlighted from the study should accompany the manuscript to support the text and offer to the audience a better understanding of the data presented.

5) The discussion section is confusing. Authors display results of the study but significant and non-significant data are not distinguished, hereditary risk factors for the next generations are not clearly defined, the prognostic value of their results is not supported.

6. PLOS authors have the option to publish the peer review history of their article (what does this mean?). If published, this will include your full peer review and any attached files.

Reviewer #1: **Yes: **Argyro Sgourou

---

## [Author Response · Author response to Decision Letter 0]

7 Aug 2020

Journal Requirements:

 : Thank you, I have checked to ensure that the journal requirements are met.

2. Please include additional information regarding the questionnaire used in the study and ensure that you have provided sufficient details that others could replicate the analyses. For instance, if you developed a questionnaire as part of this study and it is not under a copyright more restrictive than CC-BY, please include a copy, in both the original language and English, as Supporting Information.

: We added that both Thai and English versions of the questionnaire that we developed for this particular study were used.

3. Please state in your methods section the participant recruitment date.

: The information and method of selection is described in the “Data collection procedure” section. In brief, a simple random method was used to select the schools, and all students graded 4-6 in selected schools were eligible and invited to participate in the study after obtaining consent from their parents.

4. We note you have included a table to which you do not refer in the text of your manuscript. Please ensure that you refer to Table 7 in your text; if accepted, production will need this reference to link the reader to the Table.

: Thank you, this has been clarified on page 6.

Comments to the Author

Reviewer #1: The manuscript by Tawatchai Apidechkul et al. “Epi-genetics of thalassemia among the hill tribe population in Thailand” presents results of a genetic screening, aiming to estimate the prevalence of various forms of thalassemia among the six main hill tribe (minority) populations in Thailand. The issue of management of hemoglobinopathies patients is still apparent despite attempts to be controlled with prenatal testing and it is even nowadays one of the most common genetic diseases. The prevalence of all forms of thalassemia is high in countries with diverse ethnicity, high immigrant rate and in minority societies where consanguineous marriages are widely in practice. The early onset of the disease (few months after birth) and the application of expensive therapeutic protocols, including blood transfusions and chelating agents, adds an unendurable financial burden to healthcare systems.

This study appears to have been carefully performed and conclusions are relevant to the main scope of the study, the mandatory establishment of genetic counseling and prenatal diagnosis for hemoglobin disorders, especially in minority societies such as the hill tribe population in Thailand.

However, there are a number of major issues in the manuscript that need to be addressed before resubmission of revised version:

1) The term “epigenetics” in the title is irrelevant to the subject of this study. Epigenetics refers to biochemical alterations of the histone proteins or genome (DNA) or non-coding RNAs that differentially regulate gene expression levels.

: Thank you for the comment. As my training background is in “epidemiology”, which is the study and analysis of the distribution, patterns and determinants of health and disease conditions in defined populations, and our work focuses on the genetics of thalassemia by using PCR and HPLC, we used the term of “Epi-genetics”. 

However, the reviewer did not provide alternate phrasing, and we have changed the title to “Epidemiology of thalassemia among the hill tribe population in Thailand: a cross sectional study”, which seems to well reflect the content and STROBE guideline.

2) The manuscript needs an extensive grammatical and syntactical editing. In its present form does not meet the criteria for publishing.

: The manuscript has been revised twice by the American Journal Experts with the code 07A3-ECE2-5B78-98D6-6F9F . 

we hope that the language has been improved.

3) The authors need to be as precise as possible about how they did the genetic analysis

: Thank you for the comment. We have added detail to the methods section.

a)The equation in the paragraph “sample size”, although its reference is given, its terms are not (even briefly) described.

: Thank you for the comment; this has been clarified. 

b) The whole section of “materials and methods” is insufficient. The lab tests must be described and results obtained by each method performed, must be stated. Differences detected between lab methods and presented in the discussion section, must be explained and substantiated.

: The details of the laboratory methods have been added to the “Laboratory test” section. The results in each step or method are clearly presented in Tables 4-7, including elaboration of the main findings in each section. 

: In the discussion, we added to present the overall results and comparisons with other studies and populations, with possible reasons for differences. We also added information per your comments.

4) A figure presenting α-type and β-type globin genes with specific mutations highlighted from the study should accompany the manuscript to support the text and offer to the audience a better understanding of the data presented.

: Relevant information has been added to the introduction section (first paragraph)

5) The discussion section is confusing. Authors display results of the study but significant and non-significant data are not distinguished, hereditary risk factors for the next generations are not clearly defined, the prognostic value of their results is not supported.

: Thank you for the comment. However, the main aim of the study was to estimate the prevalence and the major form of thalassemia in each specific group of population, including sex and tribe. Therefore, for the main result of the study, we used the laboratory findings to present the situation of thalassemia among the hill tribe, which has never been reported. This is basic and pioneering scientific information regarding thalassemia in these populations. Detecting the risk factors for the next generation was not the main goal and is not to common because thalassemia is genetic-related disease. 

We use the term of ‘epidemiology’ to refer to description (Descriptive epidemiology) the characteristics of the person (sex and tribe) and the proportion of different forms of thalassemia, which is a descriptive epidemiology focus.

However, in Tables 4 and 5, the chi-square test is presented to detect differences in the proportion on the OF and DCIP results among patients with different characteristics. The results indicate that females are more likely to have thalassemia than males and that those whose grandfather (father of mother) and grandmother (mother of mother) are of the same tribe were more likely to have thalassemia than those whose grandfather (father of mother) and grandmother (mother of mother) are from different tribes. 

Moreover, α-thalassemia 1 was found at a higher rate in females than males, though β-thalassemia was higher among males than females. 

We have heavily reviewed all the literature of different sources of medical papers, but unfortunately there is no articles regarding these points. 

6. PLOS authors have the option to publish the peer review history of their article (what does this mean?). If published, this will include your full peer review and any attached files.

Kind regards,

TK

Assist Prof. Dr. Tawatchai Apidechkul, MSc (Infectious Epidemiology), PhD (Epidemiology)

Deputy Dean, School of Health Science, MFU

Director, Center of Excellence of Hill Tribe Health Research, WHO-CC 

Former Hubert H Humphrey Fellow (2013-2014), Emory University

Global Health Delivery Intensive (Harvard School of Public Health)

---

## [Decision Letter · Decision Letter 1]

2 Dec 2020

PONE-D-20-14674R1

Epidemiology of thalassemia among the hill tribe population in Thailand

PLOS ONE

Dear Dr. Apidechkul,

Thank you for submitting your manuscript to PLOS ONE. After careful consideration, we feel that it has merit but does not fully meet PLOS ONE’s publication criteria as it currently stands. Therefore, we invite you to submit a revised version of the manuscript that addresses the points raised during the review process.

More in details, we encourage you to address all comments raised by reviewers #1, #3 and #4, particularly those regarding the Material and Methods Section.

We look forward to receiving your revised manuscript.

Kind regards,

Michela Grosso, Ph.D.

Academic Editor

PLOS ONE

Reviewers' comments:

Reviewer's Responses to Questions

**Comments to the Author**

1. If the authors have adequately addressed your comments raised in a previous round of review and you feel that this manuscript is now acceptable for publication, you may indicate that here to bypass the “Comments to the Author” section, enter your conflict of interest statement in the “Confidential to Editor” section, and submit your "Accept" recommendation.

Reviewer #1: (No Response)

Reviewer #2: All comments have been addressed

Reviewer #3: (No Response)

Reviewer #4: (No Response)

2. Is the manuscript technically sound, and do the data support the conclusions?

Reviewer #1: Partly

Reviewer #3: Partly

Reviewer #4: Yes

3. Has the statistical analysis been performed appropriately and rigorously? 

Reviewer #1: Yes

Reviewer #3: Yes

Reviewer #4: Yes

4. Have the authors made all data underlying the findings in their manuscript fully available?

Reviewer #1: Yes

Reviewer #3: Yes

Reviewer #4: Yes

5. Is the manuscript presented in an intelligible fashion and written in standard English?

Reviewer #1: No

Reviewer #3: Yes

Reviewer #4: Yes

6. Review Comments to the Author

Reviewer #1: • The title of the study absolutely suits the text.

• The manuscript has been edited by an expert as authors claim, however, the text flow is tedious, specific points are still difficult for native speakers to follow, the analysis of the results are greatly extended, but with little necessity. The above mentioned issues reduce the readership for the manuscript and does not meet the criteria for publishing in PLOS ONE journal. Nevertheless, genetic analysis of existing minorities worldwide should gain attention due to high frequencies of pathological alleles observed among them.

• Carriers of pathological thalassemia alleles, should be identified as heterozygotes or double heterozygotes in case of the co-existence of two distinct alleles for thalassemia traits.

• In materials and methods the research outline should be presented clearly. The questionnaires cannot be considered as ”research instruments”.

• A list of abbreviations is needed.

• It would be more interesting to reduce the text in terms of appearing as a “letter to the editor” (than original article), with sharp analysis of results and clear definition of the consequent needs for constant genetic counseling and prenatal diagnosis for hemoglobin disorders in these minorities (hill tribe population of Thailand).

Reviewer #3: 1-I agree with all first reviewer comments, additionally, I have the following comments:

2-Materials and methods still need additions and clarifications ,starting from the manufacturer of all kits used[ KKU-OF reagent KKU-DCIP-Clear reagent kit ] instruments as well (e.g what was the brand name of qPCR used and manufacturers, HPLC machine( Variant II ??) .

3-OF and DCIP tests needs more clarifications ( as supplement to the paper probably ). These tests are not widely spread and the method of action needs to be mentioned. How the authors considered each of these tests results as positive or negative ? any references ? these are manual kits? ,Spectrometer used ? which wavelength ? Please write concise details about each tests to be more >

4- Since the above tests were the screening tests , we need to know the implication of each test . DCIP is mentioned by the authors to measure unstable Hbs, while no such Hb was mentioned in the results .I think HbE ais detected by this kit too.

5-Screening was performed using both tests as mentioned but figure -1- states OF or DCIP. Please clarify .

6- Did the authors followed previous guidelines regarding hemoglobinopathies screening and have chosen to organised their plan accordingly ? I am asking because I cant understand why after double positive tests, HPLC test was performed ? Why not after single positive tests for example ?

7- Abbreviations in Fig 1 need standardization . What did the author mean with A2A?

8-How could the authors identify HbBarts ?

9-What were the types of beta genes mutations reported in this study ? The authors have mentioned beta 0 and beta + mutation??

10-What was the abnormal Hb type detected (mentioned in Fig 1 ) please ?HPLC usually identify Hb structural variants like HbD, HbE ,HbO ,HbS,.......

11-What was the distribution of alfa-thalassemia SEA and Thai deletion among alfa thal minor cases detected ?

12-In Fig -1- the authors mentioned that qPCR was used to detect alfa thal mutations, while they have reported beta thal mutation and HbE as well ( beta structural variant ) .There are errors in the results of qPCR, some of them are mentioned in HPLC results too. Please re-arrange the whole Fig 1 .

13-i wished to see a comparison of alfa and beta thal mutations reported with surrounded districts or populations , or even the tribes included in the current study.

In conclusion ,

Material and methods needs a good revision, and results accordingly.

Reviewer #4: Comments for the author

This paper demonstrated the epidemiology of thalassemia among the hill tribe population in Thailand. The study showed differences in the prevalence of thalassemia and hemoglobinopathies among different tribes. The author also reported the relation between positive OF tests and the same-tribe grandparents. These are interesting findings reflecting the association between consanguineous marriage and thalassemia disease. I have the following comments on this paper:

1.The authors should clarify the remaining 32 patients who have positive OF tests with normal hemoglobin typing by excluding other conditions that may cause false-positive OF tests. It is interesting if the author can further test for alpha-thalassemia-2 (if not, please add this issue in the limitation part).

2.The author should demonstrate the “magnitude of the relation between thalassemia and the same-tribe grandparents” by using other statistical methods, for example, the logistic regression to show odds ratio (OR) and 95% CI.

7. PLOS authors have the option to publish the peer review history of their article (what does this mean?). If published, this will include your full peer review and any attached files.

Reviewer #1: **Yes: **Argyro Sgourou, SST HOU

---

## [Author Response · Author response to Decision Letter 1]

5 Jan 2021

Response to reviewers’ comments

Dear editor and reviewers,

In this revised version we have revised and confirmed in whole section of laboratory results with consulting experts in the field. We also made more analysis according to the comment of reviewer no.4. We did improvement in telling the story in whole text particularly in first paragraph, and many paragraphs in the method and results sections. It’s also improved by the native speakers to make sure that it could be good enough for the journal. 

I can confirm that all the points suggested from revisers have been carefully improved. 

I do very hope very hope that you happy in this version.

TK

Reviewer #1: • The title of the study absolutely suits the text.

: Thank you.

• The manuscript has been edited by an expert as authors claim, however, the text flow is tedious, specific points are still difficult for native speakers to follow, the analysis of the results are greatly extended, but with little necessity. The above mentioned issues reduce the readership for the manuscript and does not meet the criteria for publishing in PLOS ONE journal. Nevertheless, genetic analysis of existing minorities worldwide should gain attention due to high frequencies of pathological alleles observed among them.

: The whole paper has been revised and improved by both authors, and the native speakers.

: We thank you for your comment here. Even it is very difficult for us to demonstrate an excellent English, but after carful revisions many times, we do very hope that you will happy in this version.

• Carriers of pathological thalassemia alleles, should be identified as heterozygotes or double heterozygotes in case of the co-existence of two distinct alleles for thalassemia traits.

: Thank you for the great comment. We have revised all terminologies and presented only essential forms in the whole text. However, if I did not make clear to response your comment, please clarify more. Thank you so much. 

• In materials and methods the research outline should be presented clearly. The questionnaires cannot be considered as “research instruments”.

: Thank you, we agree with and it’s deleted. We have added laboratory procedures. 

• A list of abbreviations is needed.

: We do not sure that the style of Ploseone allows us to present a list of abbreviation separately. However, we have double-checked all abbreviations are placed properly in whole text. 

• It would be more interesting to reduce the text in terms of appearing as a “letter to the editor” (than original article), with sharp analysis of results and clear definition of the consequent needs for constant genetic counseling and prenatal diagnosis for hemoglobin disorders in these minorities (hill tribe population of Thailand).

: Thank you for great comment. We have made improvement in whole text particularly in the first paragraph of introduction.

Reviewer #3: 

1-I agree with all first reviewer comments, additionally, I have the following comments:

: We have carefully revised and improved all points on the first reviewers’ comments

2-Materials and methods still need additions and clarifications, starting from the manufacturer of all kits used [ KKU-OF reagent KKU-DCIP-Clear reagent kit] instruments as well (e.g what was the brand name of qPCR used and manufacturers, HPLC machine ( Variant II ??) .

: Thank you, all essential information have been placed. Page No.5, 

: Variant-II was used for HPLC. Page 5

3-OF and DCIP tests needs more clarifications (as supplement to the paper probably). These tests are not widely spread and the method of action needs to be mentioned. How the authors considered each of these tests results as positive or negative ? any references ? these are manual kits? ,Spectrometer used ? which wavelength ? Please write concise details about each tests to be more.

: Thank you so much for the great comment. We have put extended information relevant to these methods. 

: We have added detail of these procedures at page 5 with references.

4- Since the above tests were the screening tests, we need to know the implication of each test. DCIP is mentioned by the authors to measure unstable Hbs, while no such Hb was mentioned in the results. I think HbE ais detected by this kit too.

: Thank you for the comment. At the stage of using either OF or DCIP, we did not intend to make interpretation since there were screening tests. 

: Bascially, OF test intends to screen for alpha-thalassemia-1 and beta-thalssemia, while DCIP is screened for Hb E. Then, in our study, after having a positive of either one or both were confirmed by HPLC and PCR before making interpretation. 

:If OF+ only, it’s suspected to be alpha-thalassemia and/ or betha-thalssemia, while having DCIP+ only, it’s suspected to be Hb E trait. If having positive in both tests, it could be suspected Hb E with or without alpha-thalassemia, and/or b-thalassemia. 

5-Screening was performed using both tests as mentioned but figure -1- states OF or DCIP. Please clarify.

: Sorry for the mistake, it’s correct. Specimens were tested by both tests.

6- Did the authors followed previous guidelines regarding hemoglobinopathies screening and have chosen to organised their plan accordingly ? I am asking because I cant understand why after double positive tests, HPLC test was performed ? Why not after single positive tests for example ?

: Positive to either OF or DCIP, or both OF and DCIP were tested by HPL and PCR. 

7- Abbreviations in Fig 1 need standardization. What did the author mean with A2A?

: Thank you, there have been made clarification all abbreviations.

: HbA which is adult hemoglobin, hemoglobin A1 or �2�2. 

: HbA is the most common adult form of hemoglobin and exist as tetramer containing two alpha subunits and two beta subunits (�2�2)

: HbA2 is a less common adult form of hemoglobin and is composed of two alpha and two delta.

: A2A is a normal variant of hemoglobin A that consists of two alpha and two delta chains (�2�2) and is found at low levels in normal human blood.

8-How could the authors identify HbBarts ?

: By doing Hb typing in the HPLC method, those who have presented the low Hb A2 level due to having Hb H (�4) or Hb Bart’s (�4) which is few of �-globin then the �-globin and �-globin remain a large portion and catching each other to obtain 4 lines. Then while we doing HPLC, the band of Hb Barts is presented.

9-What were the types of beta genes mutations reported in this study ? The authors have mentioned beta 0 and beta + mutation??

: It could be (β0/β) or (β+/β)

: However, in our current method, we could not identify in specific genes mustation. Sorry.

10-What was the abnormal Hb type detected (mentioned in Fig 1 ) please ?HPLC usually identify Hb structural variants like HbD, HbE ,HbO ,HbS,.......

: Thank you for the notice. 

: A case of abnormal Hb type was detected by HPLC with detail; MCV=73.4lf, positive OF, positive DCIP, 46.0%HbAo, 3.0%HbA2/E, 0.5%HbF, A2A Hb typing with abnormal Hb, and negative for alpha-thalassemia (SEA and Thai deletion). 

11-What was the distribution of alfa-thalassemia SEA and Thai deletion among alfa thal minor cases detected?

: Only alpha-thalassemia 1 trait (SEA deletion) was found, data presented in table 6 and 7.

12-In Fig -1- the authors mentioned that qPCR was used to detect alfa thal mutations, while they have reported beta thal mutation and HbE as well (beta structural variant) .There are errors in the results of qPCR, some of them are mentioned in HPLC results too. Please re-arrange the whole Fig 1 .

: Thank you for great comment, we have made revised in whole structure in figure no. 1, and also in the context.

13-i wished to see a comparison of alfa and beta thal mutations reported with surrounded districts or populations, or even the tribes included in the current study.

: It’s reported in the reference no. 21 and 28. There was very few publications on the hill tribe people in Thailand. One more study reported on beta thalassemia among the hill tribe school children (Ref. No. 29). 

In conclusion ,

Material and methods needs a good revision, and results accordingly.

: Thank you very much for the great comments. We have revised improved a large scale particularly in extending the content of OF and DCIP including other information as comments. 

Reviewer #4: Comments for the author

This paper demonstrated the epidemiology of thalassemia among the hill tribe population in Thailand. The study showed differences in the prevalence of thalassemia and hemoglobinopathies among different tribes. The author also reported the relation between positive OF tests and the same-tribe grandparents. These are interesting findings reflecting the association between consanguineous marriage and thalassemia disease. I have the following comments on this paper:

1.The authors should clarify the remaining 32 patients who have positive OF tests with normal hemoglobin typing by excluding other conditions that may cause false-positive OF tests. It is interesting if the author can further test for alpha-thalassemia-2 (if not, please add this issue in the limitation part).

: I have careful looked into your point, unfortunately I do not see the particular point. From Table 6, 117 of 1,200 cases have OF+/DCIP+/ or OF+ and DCIP+.

: In total, 97 cases of OF+; 79 case A2A, 1 case EE, 15 cases EA, 1 case A2A Barts H, and 1 case A2A with abnormal Hb.

: another 4 cases of A2A have OF- but DCIP+. 

: Therefore, all OF positives have presented some form of Hb disdorder. 

: 30 cases of alpha thalassemia 1 (SEA deletion), 29 cases have A2A (Hb A2<3.5%), and one case has A2A Bart’s H.

: However, if I missed the point please kindly let me know. 

2.The author should demonstrate the “magnitude of the relation between thalassemia and the same-tribe grandparents” by using other statistical methods, for example, the logistic regression to show odds ratio (OR) and 95% CI.

: Thank you, we have used logistic regression to see the magnitude of having OF+ or DCIP+ or positive both OF and DCIP in different models. We have added the information into the result (Page 18-19) and discussion sections.

Thank you,

TK

---

## [Decision Letter · Decision Letter 2]

26 Jan 2021

Epidemiology of thalassemia among the hill tribe population in Thailand

PONE-D-20-14674R2

Dear Dr. Apidechkul,

We’re pleased to inform you that your manuscript has been judged scientifically suitable for publication and will be formally accepted for publication once it meets all outstanding technical requirements.

Kind regards,

Michela Grosso, Ph.D.

Academic Editor

PLOS ONE

Additional Editor Comments (optional):

Reviewers' comments:

Reviewer's Responses to Questions

**Comments to the Author**

1. If the authors have adequately addressed your comments raised in a previous round of review and you feel that this manuscript is now acceptable for publication, you may indicate that here to bypass the “Comments to the Author” section, enter your conflict of interest statement in the “Confidential to Editor” section, and submit your "Accept" recommendation.

Reviewer #3: All comments have been addressed

Reviewer #4: All comments have been addressed

2. Is the manuscript technically sound, and do the data support the conclusions?

Reviewer #3: Yes

Reviewer #4: Yes

3. Has the statistical analysis been performed appropriately and rigorously? 

Reviewer #3: Yes

Reviewer #4: Yes

4. Have the authors made all data underlying the findings in their manuscript fully available?

Reviewer #3: Yes

Reviewer #4: Yes

5. Is the manuscript presented in an intelligible fashion and written in standard English?

Reviewer #3: Yes

Reviewer #4: Yes

6. Review Comments to the Author

Reviewer #3: I appreciate the authors efforts in correcting the whole work.One minor comment please; the laboratory section in MM needs language editing please .

Thank you

Reviewer #4: This is a large epidemiologic study of thalassemia among the hill tribe population. The author showed differences in the prevalence of thalassemia and hemoglobinopathies among different tribes. The author demonstrated the relation between positive OF tests and the same-tribe grandparents. These findings reflected the association between consanguineous marriage and thalassemia disease. The author has addressed all comments. I have no further comment.

7. PLOS authors have the option to publish the peer review history of their article (what does this mean?). If published, this will include your full peer review and any attached files.

Reviewer #3: No

Reviewer #4: No

---

## [Editor Report · Acceptance letter]

29 Jan 2021

PONE-D-20-14674R2 

Epidemiology of thalassemia among the hill tribe population in Thailand 

Dear Dr. Apidechkul:

I'm pleased to inform you that your manuscript has been deemed suitable for publication in PLOS ONE. Congratulations! Your manuscript is now with our production department. 

Kind regards, 

on behalf of

Prof. Michela Grosso 

Academic Editor

PLOS ONE